# Implementation and Evaluation of Age-Aware Downlink Scheduling Policies in Push-Based and Pull-Based Communication

**DOI:** 10.3390/e24050673

**Published:** 2022-05-11

**Authors:** Tahir Kerem Oğuz, Elif Tuğçe Ceran, Elif Uysal, Tolga Girici

**Affiliations:** 1Department of Electrical and Electronics Engineering, Middle East Technical University, Ankara 06800, Turkey; elifce@metu.edu.tr (E.T.C.); uelif@metu.edu.tr (E.U.); 2Department of Electrical and Electronics Engineering, TOBB University of Economics and Technology, Ankara 06560, Turkey; tgirici@etu.edu.tr

**Keywords:** age of information, query age of information, wireless networks, software-defined radio, scheduling

## Abstract

As communication systems evolve to better cater to the needs of machine-type applications such as remote monitoring and networked control, advanced perspectives are required for the design of link layer protocols. The age of information (AoI) metric has firmly taken its place in the literature as a metric and tool to measure and control the data freshness demands of various applications. AoI measures the timeliness of transferred information from the point of view of the destination. In this study, we experimentally investigate AoI of multiple packet flows on a wireless multi-user link consisting of a transmitter (base station) and several receivers, implemented using software-defined radios (SDRs). We examine the performance of various scheduling policies under push-based and pull-based communication scenarios. For the push-based communication scenario, we implement age-aware scheduling policies from the literature and compare their performance with those of conventional scheduling methods. Then, we investigate the query age of information (QAoI) metric, an adaptation of the AoI concept for pull-based scenarios. We modify the former age-aware policies to propose variants that have a QAoI minimization objective. We share experimental results obtained in a simulation environment as well as on the SDR testbed.

## 1. Introduction

The advent and the fast growth of the Internet of things (IoT) has further complicated the design of communication networks, in the presence of an increase in demand in networked services catered by the fifth generation (5G) evolution of communication networks. On the one hand, machine-type communications are typically less bandwidth hungry than typical multimedia services. On the other hand, IoT flows tend to be composed of many small packets generated by large numbers of end nodes, and they may have end-to-end freshness requirements that may be challenging to satisfy with conventional link or transport layer approaches based on optimizing throughput and delay. Increasing the sampling rate of IoT nodes to respond to freshness requirements or adopting first-come-first-served service policies can cause bottlenecks on the network, resulting in a reduction in quality of service. It has been argued in recent literature that optimizing data generation, transmission, and transport with respect to higher-level metrics such as Age of Information can prevent unnecessary network load, while improving the freshness of flows. In a broader perspective, there are proposals to encapsulate the significance or the value of the transferred information to the communication problem in certain “semantic metrics” and use these in the design of algorithms and protocols in all network layers, referred to as “semantic communication” [1].

Within the set of semantic metrics, the age of information (AoI) from the receiver’s point of view is defined as the time elapsed since the generation of the newest status update that has been received by the destination [2]. AoI is gaining momentum as a key performance indicator (KPI) for machine-type communications (MTC). The primary reason for the interest in AoI is the growing demand for timely and fresh information in many emerging real-time and remote monitoring-based applications such as the Internet of things, vehicular networks, and cyber-physical systems.

AoI monitors the freshness of the entire information stream from the receiver’s point of view. Hence, it reveals further aspects of the network compared to traditional metrics such as delay or throughput. For instance, the delay metric measures the timeliness from the transmitted packet’s perspective. A low average delay does not mean a low average age in every case [3]. Continuous packet transmission policy (known as zero-wait policy in the literature) can optimize delay, but it may not provide age-optimality in the presence of FCFS (first-come-first-serve) queues [4]. Moreover, if the transmitter has an energy constraint, the inefficiency of the zero-wait policy becomes more apparent [3]. Improving the throughput alone can maximize the amount of data flow to the receiver node but may cause an overload of the queues within the network. Packets waiting in the queue result in outdated information reaching the receiver node. In this case, to reduce backlogs within queues, the packet generation rate should be decreased. However, an over-reduction of the packet generation rate would cause the receiver to be updated sporadically, which also leads to reduced AoI performance. This dilemma shows that AoI is a composite measure of both throughput and delay. For achieving optimal AoI, frequent packets must arrive regularly [5]. Consequently, solving the scheduling problem with an AoI minimization objective requires a novel formulation.

A significant portion of the AoI literature consists of studies involving push-based communication scenarios. In the push-based model, the generation of a new packet triggers the communication process. Then, the transmitter module sends the generated packet to the receiver module. The sequence of operations of the communication process proceeds from the information source to the destination. However, one of the network models often encountered in real-life scenarios is the pull-based model, where the query source requests (or queries) information from the receiver module. In this scenario, the initiator of the communication process is the query source that aims to pull information from the receiver module. The source of these queries could be users or applications that want to monitor the information source. In the pull-based network, the sequence of operations of the communication process proceeds from the destination to the source.

In this paper, we consider both push-based and pull-based status update systems and experimentally investigate the performance of several age-aware downlink scheduling policies in wireless multi-user networks. The main contribution of this study is to report one of the pioneering experimental studies of age-aware MAC layer scheduling policies. We have implemented a multi-user downlink network with a single base station and multiple receivers using software-defined radios (SDRs). This testbed implementation allowed us to examine push-based and pull-based scenarios and state-of-the-art scheduling policies. Along with the other well-known policies, we have proposed max-weight policies for different pull-based scenarios and provided extensive simulation and experimental results.

The rest of the paper is organized as follows. In Section 2, we present the related work. In Section 3, the system model is presented and the problems of minimizing the average AoI, QAoI, gmagmaild and EAoI in the network are formulated. Age-aware downlink scheduling policies are exhibited in Section 4, and the experimental setup is explained in detail in Section 5. Simulation and experimental results are presented in Section 6, and the paper is concluded in Section 7.

## 2. Related Work

There are numerous studies examining the AoI metric in the literature. The major works that stand out in the literature are those investigating the effects of different queuing types and developing scheduling policies to minimize average AoI in the network. An important concern when proposing a scheduling policy is the required computational load [5]. The work in [6] shows that a scheduling problem with an age minimization goal is an NP-hard problem in a multi-user network. In [7], age-aware scheduling policies are derived for the lossy channel case in a multi-user network. The greedy policy is inspected, and results indicate that the policy is optimal in the symmetric channel state for mean AoI minimization. In [8], the network is analyzed based on the peak-age and mean-age metrics, and a virtual-queue-based policy and an age-based policy are developed. The virtual-queue-based policy is shown to be peak-age optimal. The age-based policy is proved to be within a factor of four of optimal values for peak age as well as average age. In [5], Whittle’s index (WI) policy and max-weight (MW) policy are proposed. The lower bound for AoI that can be calculated by using the statistical information of the network is derived. Lower and upper limits of AoI performances for WI and MW policies are calculated and proven to be within a factor of four of the optimal (upper limit is at most four times higher than the lower limit). There are also learning-based approaches in the literature to find an optimal age-aware policy for multi-user networks [9,10].

In the multi-user scheduling problem, the generation procedure of the packets has a significant impact on the AoI. In the literature, sources that generate a fresh packet at every time frame are referred to as “active sources” [8]. For a system model with active sources, whenever there is a transmission, the age of the corresponding flow will be reset to its minimum possible value (one frame duration in our setup). However, many realistic scenarios may be better modeled with a packet generation that is a stochastic process. For example, [11] studied a case where the packet generation procedure is a Bernoulli stochastic process and proposed scheduling policies suitable for that system model.

The queue service policy (e.g., LCFS (last-come-first-serve), FCFS (first-come-first-serve)) also has a significant effect on AoI [12,13]. For the active source case, queuing policies become even more important since sources load the network with the highest rate available. Queue management policy determines the behavior of the queue when the new packets arrive. If the queue is managed with an LCFS policy, the freshest packet will be at the top of queue, and the first packet that leaves the queue will be the one with the most up-to-date information. In FCFS queues, a new packet is added to the bottom end of the queue. To transmit the most timely packet, all packets in front of the last inserted packet must be sent for transmission. As a result, the most up-to-date packet loses time and becomes stale waiting the transmission of other packets in the queue.

The overwhelming majority of the AoI literature to date has emphasized theoretical studies. However, there are also studies on implementation in the literature [14,15,16,17,18,19,20,21,22]. For a survey of this implementation-oriented literature, see [23]. In [14,15,16,17], the experimental setup mostly lies between the transport layer and application layer. The effects of different wireless access technologies on end-to-end TCP/IP connections were measured by [14,15,16]. Studies in [18,19,20] cover a broader range of interconnection layers and capture the performances of novel age-based MAC layer algorithms. In [18], Wi-Fi protocol is implemented on SDRs. The uplink of a wireless network is taken into consideration, and the effect of utilizing the MW scheduling policy is investigated. The work presented in [19] experimentally investigates the effects of packet management policies on the performance of networked control systems. A test environment was developed by [20] to evaluate various ALOHA-like random access protocols. In our previous work in [21], we implemented a multi-user wireless network using SDRs. We compared the AoI performances of MW and WI policies with round-robin and greedy policies.

The time-average age metric weighs information freshness of all time frames equally. However, there are many types of real-world applications where the demand for timely information varies in time. For these, minimization of time-average age may not be the most relevant objective. In the literature, various semantic metrics alter this model, placing higher emphasis on selected time frames. For example, the age of incorrect information (AoII) metric focuses on the usefulness of the information and aims to maximize the freshness of non-repetitious information. In the AoII concept, obtaining redundant information is pointless for the receiver and does not reduce AoII. The objective is to minimize the age of differing information [24,25,26].

Query age of information (QAoI) is a recent semantic metric proposed to investigate pull-based scenarios from the AoI perspective [27,28,29,30]. QAoI considers a model where the freshness of information is valuable only at query moments. These queries are sent to the receiver modules in the network. Then, the receiver modules respond to these queries with the most up-to-date information. The source of the queries can be a user or an application that needs to obtain the most up-to-date information. In [29], the pull-based scenario is discussed, and the effective age of information (EAoI) metric is presented for the multi-user system model. Query generation is modeled as an independent Bernoulli process for each receiver, and the immediate EAoI is assumed to be zero for frames without queries. For the queried frames, immediate EAoI is related to the immediate AoI of the receiver under the proactive serving assumption as a query response procedure. According to the proactive serving method, the receiver module can wait for the query response for a frame if it is expecting a packet arrival within the frame. If a packet arrives at the end of the frame, the receiver sends the information in the newly received packet as a query response. In the study, WI-based scheduling policy is proposed for the multi-user system model, and the performance of the policy is demonstrated in the simulation environment.

The work in [28] presents the query-AoI metric for a single receiver in pull-based communication. The calculation of the QAoI metric presented in this study is similar to the EAoI. However, an instantaneous serving scenario is adopted instead of the proactive serving in [29]. In addition, the transmitter module is assumed to have an energy constraint, and the presence of the energy constraint turns the problem in another direction while increasing the value of the QAoI reduction per transmission. Within the scope of the study, the permanent query (PQ) model, which is a query generation procedure that approximates the studied problem with the standard AoI problem, and the query arrival process-aware (QAPA) model, which generates queries based on periodic or stochastic processes, are examined. The optimal scheduling policy in the PQ process is the same as the optimal scheduling policy for AoI. In the case of QAPA, the scheduling policy has information on the query process (either stochastic or deterministic) and can schedule accordingly.

A continuous-time status update model is investigated in [30], where a source node submits update packets to a channel with random transmission delay, and the query source tries to pull information from the receiver module according to a stochastic arrival process. The average QAoI is defined as the average AoI measured at query instants, and the system model is examined from both AoI and QAoI perspectives. Age-aware scheduling policies do not use the information about the query process and freshness equally for all frames. On the other hand, QAoI-aware scheduling policies use additional information about the query process in the scheduling decisions. This extra information allows the scheduling policy to distribute transmission attempts more efficiently and reduce the time spent in the FCFS queue. Eventually, from the query source’s perspective, QAoI-aware policies can provide better AoI performance than AoI-aware policies.

To the best of our knowledge, this is the first work in the literature that considers practical implementation and evaluation of QAoI-aware scheduling policies. In addition, we propose and implement novel max-weight policies for the effective AoI and query-AoI system models and evaluate their performance in terms of AoI, EAoI, and QAoI, in both simulation and SDR environments. We have observed that the resulting EAoI-aware max-weight (EAoI-MW) policy has a similar EAoI performance to the WI but yields a higher network throughput. We have also observed that QAoI-aware max-weight (QAoI-MW) provides superior QAoI performance than AoI-aware policies.

## 3. System Model

We consider a wireless multi-user network where a common access point or a base station (BS) needs to send status update packets containing time-sensitive information to multiple receiver modules. Let *M* denote the total number of receivers. We also assume a discrete time system where time is divided into fixed-length frames denoted by t∈{1,…,T}. In each frame, the base station (transmitter) is allowed to activate the connection for a single receiver i∈{1,…,M}, and it cannot send packets to more than one receiver within a frame. A transmission attempt of a status update to a single user takes a constant time, which is assumed to be equal to the duration of one frame. Wireless channels between the receivers and the base station are unreliable. The state of each channel changes randomly from one time slot to the next and is modeled by a Bernoulli random variable. Channel states for each receiver are also independent of the others.

The packet generation scheme in the system follows the “active source” model. At the beginning of each frame, information sources generate new packets for each receiver, and these packets reach the BS immediately. The base station selects one of these packets for transmission and discards the others. There are no queuing-related delays between the information sources and the base station. If a receiver successfully receives a packet, the AoI of this receiver successfully drops to one since the newly formed packet at the information source reaches the receiver within a frame, without observing any delay.

In the system model, there are also query sources linked to each receiver. Each query source is independent of the other, and used to model the behavior of a real-life user or application interested in a particular time-sensitive piece of information at query instants. Query arrival frames to receivers can follow either a deterministic or stochastic pattern. When a query source requests information from a receiver, it sends a query. Then, the receiver responds to it with the latest information that the receiver holds. Query and response messages are transmitted without any errors.

The BS judiciously selects a receiver for transmission according to a stationary scheduling policy π∈Π represented by $a_{i}\left(t\right)$, for all i∈{1,…,M} and t∈{1,…,T}. If the receiver *i* is selected for transmission in frame *t*, then $a_{i}\left(t\right)$ will be equal to one. Otherwise, $a_{i}\left(t\right)$ will be equal to zero. Evaluation of $a_{i}\left(t\right)$ is given in (Equation 1).
(1)$$a_{i}\left(t\right)=\left\{\begin{array}{cc} 1 & \;\mathrm{if}\;\mathrm{the}\;\mathrm{receiver}\;i\;\mathrm{is}\;\mathrm{selected},\\ 0 & \;\mathrm{otherwise}\;. \end{array}\right.$$

If a successful transmission occurs, the base station is informed over an error-free channel in the same frame. By utilizing this knowledge, scheduling policy can keep track of the AoI of the receivers. Similarly, $c_{i}\left(t\right)$ is a binary variable indicating the random channel state of receiver *i* at frame *t*. If the channel status of the receiver *i* is ON, then the successful transmission can be made at frame *t*, and $c_{i}\left(t\right)$ will be equal to one. Otherwise, if the channel is not available for transmission, $c_{i}\left(t\right)$ will be equal to zero. We assume $c_{i}\left(t\right)$ is an independent and Bernoulli-distributed random variable and the probability of successful transmission (i.e., reliability) is $p_{i}$, for all i∈{1,…,M}. Evaluation of $c_{i}\left(t\right)$ is given in (Equation 2).
(2)$$c_{i}\left(t\right)=\left\{\begin{array}{cc} 1 & \;\mathrm{if}\;\mathrm{the}\;\mathrm{channel}\;\mathrm{is}\;\mathrm{ON}\;,\\ 0 & \;\mathrm{if}\;\mathrm{the}\;\mathrm{channel}\;\mathrm{is}\;\mathrm{OFF}\;. \end{array}\right.$$

To have a successful transmission in frame *t*, the receiver *i* must be selected for transmission, and the channel status of that receiver must be available for transmission. Let $u_{i}\left(t\right)$ denote the overall result of the transmission to receiver *i* at frame *t* (Equation 3). Evaluation of $u_{i}\left(t\right)$ is given in (Equation 3).
(3)$$u_{i}\left(t\right)=\left\{\begin{array}{cc} 1 & \;\mathrm{if}\;c_{i}\left(t\right)a_{i}\left(t\right)=1,\\ 0 & \;\mathrm{otherwise}\;. \end{array}\right.$$
We also define $f_{i}\left(t\right)$ as complementary of $u_{i}\left(t\right)$ for simplification of some equations throughout the paper, that is, $f_{i}\left(t\right)=1-u_{i}\left(t\right)$.

The instantaneous AoI of receiver *i* at the beginning of the *t*th frame is denoted by $\Delta_{i}\left(t\right)$. Note that $\Delta_{i}\left(t\right)$ drops to one if the transmission to receiver *i* succeeds and increases by 1 if receiver *i* is not selected for transmission or fails to successfully receive a packet. Evaluation of $\Delta_{i}\left(t\right)$ is given in (Equation 4).
(4)$$\Delta_{i}\left(t+1\right)=\left\{\begin{array}{cc} 1 & \;\mathrm{if}\;u_{i}\left(t\right)=1\\ \Delta_{i}\left(t\right)+1 & \;\mathrm{otherwise}\; \end{array}\right.$$

$d_{i}\left(t\right)$ indicates the query presence. If a query arrives to the receiver *i* at frame *t*, $d_{i}\left(t\right)$ will be equal to one. Otherwise, $d_{i}\left(t\right)$ will be equal to zero. Evaluation of $d_{i}\left(t\right)$ is given in (Equation 5).
(5)$$d_{i}\left(t\right)=\left\{\begin{array}{cc} 1 & \;\mathrm{if}\;a\;\mathrm{query}\;\mathrm{arrives}\;\mathrm{to}\;\mathrm{the}\;\mathrm{receiver}\;\\ 0 & \;\mathrm{otherwise}\; \end{array}\right.$$

The instantaneous query age of receiver *i* at the beginning of the *t*th frame is $\Delta_{q_{i}}\left(t\right)$. The evaluation of the $\Delta_{q_{i}}\left(t\right)$ varies with the adopted query response scenario within the system model. In the scope of this study, we assume that query arrival to the receiver and receiver’s response will happen at the beginning of the frame. We denote this query response scenario as the "instantaneous serving" scenario. Evaluation of $\Delta_{q_{i}}\left(t\right)$ for instantaneous serving scenario is given in (Equation 6).
(6)$$\Delta_{q_{i}}\left(t\right)=d_{i}\left(t\right)\Delta_{i}\left(t\right)$$

An alternative query response scenario called “proactive serving” is defined in the literature in [29]. In proactive serving, the response to the query may be delayed by at most one frame. The purpose of this delay is to put the newest information into the query if the receiver acquires a packet within the queried frame. Nevertheless, unless stated otherwise, the instantaneous serving strategy will be adopted throughout this study. The overall system model is illustrated in Figure 1.

Next, we formally define the AoI and QAoI minimization problems in Section 3.1 and Section 3.2, respectively.

### 3.1. AoI Minimization Problem

The analytical expressions for the AoI minimization problem have been previously studied in [5]. The objective of the scheduling policy is to minimize the average AoI in the network. Average AoI is calculated for *M* receivers across *T* frames. The objective is to find a stationary scheduling policy π∈Π that minimizes the long-term average AoI, which is defined in (Equation 7).
(7)$$\min_{\pi \in \Pi }\lim_{T\to \infty }E\left[J_{A}\left(\pi \right)\right],\;\mathrm{where}\;J_{A}\left(\pi \right)=\frac{1}{TM}\sum_{t=1}^{T}\sum_{i=1}^{M}\Delta_{i}\left(t\right)$$

### 3.2. QAoI Minimization Problem

For the QAoI problem, the main objective of the scheduling policy is to minimize the average age of the query sources in the network. This problem differs from the AoI problem since the query sources do not require fresh data at every instant but only at the queried frames. The difference between the two problem statements has also been previously investigated in [28,30].

There are two major approaches in the literature to calculate the average ages of the users at query instants in pull-based communication systems. In the first approach, the sum of the ages at query instants is divided by the total number of frames. This method is followed by [27,29] to develop age-aware scheduling policies, and the metric is called *effective age of information (EAoI)*. Note that [28] also follows a similar approach in the discounted setting for single-user pull-based communication.

The objective function obtained by utilizing this approach is given in (Equation 8).
(8)$$\min_{\pi \in \Pi }\lim_{T\to \infty }E\left[J_{E}\left(\pi \right)\right],\;\mathrm{where}\;J_{E}\left(\pi \right)=\frac{1}{TM}\left[\sum_{i=1}^{M}\sum_{t=1}^{T}\Delta_{q_{i}}\left(t\right)\right]$$

The second approach divides the sum of all query ages by the total number of query arrivals. This approached is used by [30] for the average query age calculation. The objective function obtained by utilizing this approach is given in (Equation 9).
(9)$$\min_{\pi \in \Pi }\lim_{T\to \infty }E\left[J_{Q}\left(\pi \right)\right],\;\mathrm{where}\;J_{Q}\left(\pi \right)=\frac{1}{M}\sum_{i=1}^{M}\frac{1}{N_{i}\left(T\right)}\left[\sum_{t=1}^{T}\Delta_{q_{i}}\left(t\right)\right],$$
where $N_{i}\left(T\right)$ denotes the total number of queries arrived at receiver *i* throughout *T* frames.

Throughout this study, we refer to the metric aligned with the first approach as the *effective age of information (EAoI)*, following its definition in [27]. We call the metric evaluated with the second approach the *query age of information (QAoI)*.

In the average EAoI calculation, the query age of the frames for which the query is not present is taken as zero and included in the average. This calculation method may lead to misleading results for measuring the average AoI of the query sources. This is because even if the AoI of a rarely queried receiver is very high at the time of query, it remains low on average due to the inclusion of unqueried frames. Similarly, for a frequently queried receiver, since the number of unqueried frames is low, the number of zeros included in the calculation of the average EAoI will be low. Therefore, the EAoI of this receiver will tend to be higher than the rarely queried receiver. The effect of the scheduling policy becomes less apparent as the query frequency decreases. Therefore, to measure the performance of scheduling policies, comparing average EAoI values of two individual systems with different query arrival frequencies would provide inconsistent results. When the same problem is analyzed from the QAoI perspective, the effect of the scheduling policy becomes more decisive, as the unqueried frames are discarded in the average query age calculation.

To examine the QAoI problem, we first consider the case where the query generation is an independent Bernoulli process. Note that [28,30] indicates that, to see a difference between QAoI and AoI metrics, the query arrival process must be non-stationary. For the Bernoulli query arrival case, the QAoI problem converges to the AoI problem. On the other hand, EAoI can yield results different than AoI even under the Bernoulli-arrival scheme.

## 4. Age-Aware Downlink Scheduling Policies

In this section, we define scheduling policies to minimize age-aware metrics. We describe AoI-aware, EAoI-aware, and QAoI-aware scheduling policies in Section 4.1, Section 4.2, Section 4.3, respectively.

### 4.1. Scheduling Policies for AoI Minimization

AoI-aware scheduling policies have been previously investigated in [5,7,8,10]. The WI and MW policies proposed in [5,7] utilize the instantaneous ages of the receivers and the reliabilities of the corresponding links to calculate the expected costs $\left\{C_{i}\right\}$ associated with each receiver. To maximize the cost reduction, the scheduling policy selects the receiver with the highest $C_{i}$ at each frame.

The max-weight policy is an adaptation of the Lyapunov optimization technique to the AoI minimization problem. Lyapunov optimization provides a method for penalty minimization while maintaining the queue stability [31]. The objective of the MW policy is to minimize Lyapunov drift in the network with the appropriate scheduling decision. Lyapunov drift measures the expected cost increase between two consecutive frames. In each frame, the policy calculates the expected Lyapunov drift of the receivers. Then, the policy selects the receiver with the highest Lyapunov drift. With this decision, the policy aims to minimize the overall cost. The calculation of expected costs for the MW policy is given in (Equation 10). At each frame, the scheduling policy selects the receiver with the highest $C_{i}$.
(10)$$C_{i}\left(\Delta_{i}\left(t\right)\right)=p_{i}\Delta_{i}\left(t\right)\left(\Delta_{i}\left(t\right)+2\right)$$

The WI policy has been presented in [5,7,10] by formulating the AoI minimization problem in (Equation 7) as a restless multi-armed bandit (MAB) problem. The MAB problem in general aims to optimize the reward in an unknown environment through a series of trials where the decision-maker can activate only one of the arms and each arm has an immediate reward (or penalty for the minimization problem case) associated with it. The closed-form costs (indexes) for the WIP are given in (Equation 11). At each frame, the scheduling policy transmits to the receiver with the highest $C_{i}$.
(11)$$C_{i}\left(\Delta_{i}\left(t\right)\right)=p_{i}\Delta_{i}\left(t\right)\left[\Delta_{i}\left(t\right)+\frac{2-p_{i}}{p_{i}}\right]$$

In our study, we implement AoI-aware MW and WI policies on the USRP testbed and compare their performances with round-robin and greedy policies.

### 4.2. Scheduling Policies for EAoI Minimization

In the USRP testbed, we implement and evaluate the performance of the EAoI-aware WI policy that was previously proposed in [29]. In addition, we propose an EAoI-aware MW policy by modifying the AoI-aware max-weight policy previously proposed in [5] and compare their performances.

EAoI-aware WI in [29] is given in (Equation 12). In each frame, the policy chooses the receiver with the highest $C_{i}$.
(12)$$C_{i}\left(t\right)=q_{i}\left(p_{i}\Delta_{i}\left(t\right)+2\right)\left(\Delta_{i}\left(t\right)-1\right)$$

We can derive the max-weight policy for the pull-based instantaneous serving scenario: First, we calculate the Lyapunov drift of the instantaneous EAoI’s between consecutive frames. Then, in line with [5], we select quadratic Lyapunov function to calculate the Lyapunov drift.

**Lemma** **1.***In each frame, EAoI-MW policy selects the receiver with highest $C_{i}\left(t\right)$, which can be computed as in*(Equation 13).
(13)$$C_{i}\left(t\right)=q_{i}p_{i}\left(\Delta_{i}^{2}\left(t\right)+2\Delta_{i}\left(t\right)\right).$$

Derivation of the EAoI-MW Policy can be found in Appendix A.

### 4.3. Scheduling Policies for QAoI Minimization

For the QAoI metric investigation, we evaluate the cases where the query arrival process forms a Markov chain. Within the Markov chain, we designate one state as the “Query” state and other states as “non-query” states. When the current state of the Markov chain reaches the query state, a query arrives.

For QAoI minimization, we propose a max-weight-based scheduling policy, following similar steps as in [5]. To adapt this policy to the QAoI model, we utilize the main features of the Markov chain, which determines the query process. In the first step, we calculate the future AoI $\Delta_{i}\left(t+K\right)$ in terms of current AoI $\Delta_{i}\left(t\right)$. The evaluation of AoI between consecutive frames is given in (Equation 14).
(14)$$\begin{array}{cc} \Delta_{i}\left(t+1\right)=\; & u_{i}\left(t\right)+\left(1-u_{i}\left(t\right)\right)\left(\Delta_{i}\left(t\right)+1\right)\\ =\; & a_{i}\left(t\right)c_{i}\left(t\right)+\left(1-a_{i}\left(t\right)c_{i}\left(t\right)\right)\left(\Delta_{i}\left(t\right)+1\right)\\ =\; & 1+f_{i}\left(t\right)\Delta_{i}\left(t\right) \end{array}$$
Repeating this approach multiple times enables us to obtain the future AoI in terms of current AoI. The result is given in (Equation 15).
(15)$$\begin{array}{cc} \Delta_{i}\left(t+1\right)=\; & 1+f_{i}\left(t\right)\Delta_{i}\left(t\right)\\ \Delta_{i}\left(t+2\right)=\; & 1+f_{i}\left(t+1\right)+f_{i}\left(t+1\right)f_{i}\left(t\right)\Delta_{i}\left(t\right)\\ \Delta_{i}\left(t+3\right)=\; & 1+f_{i}\left(t+2\right)+f_{i}\left(t+2\right)f_{i}\left(t+1\right)+f_{i}\left(t+2\right)f_{i}\left(t+1\right)f_{i}\left(t\right)\Delta_{i}\left(t\right) \end{array}$$

In the following equations, we indicate the future time frames as $\hat{t}$. Although it may lead to suboptimal results, for computational convenience, we assume that future decisions $a_{i}\left(\hat{t}\right)$ are independent variables and stationary through time with a fixed expected value. Based on this assumption, we can argue that $f_{i}\left(\hat{t}\right)$ is also stationary. Therefore, we define $f_{i}$ as the stationary version of the $f_{i}\left(\hat{t}\right)$ as shown in Equation (Equation 16).
(16)$$E\left[f_{i}\left(\hat{t}\right)\right]=E\left[f_{i}\left(t+1\right)\right]=E\left[f_{i}\left(t+2\right)\right]=E\left[f_{i}\left(t+K\right)\right]=f_{i}$$
Then, we define the closed-form version the future AoI with current AoI in (Equation 17).
(17)$$\Delta_{i}\left(t+K\right)=\left[\sum_{k=1}^{K}f_{i}^{k-1}\right]+f_{i}^{K-1}f_{i}\left(t\right)\Delta_{i}\left(t\right)$$
To simplify the notation, we define $F_{s}\left(K\right)$ and $F_{m}\left(K\right)$ as in (Equation 18) and (Equation 19).
(18)$$F_{s}\left(K\right)=\sum_{k=1}^{K}f_{i}^{k-1}$$
(19)$$F_{m}\left(K\right)=f_{i}^{K-1}$$
We then rewrite the simplified version of Equation (Equation 17) in Equation (Equation 20).
(20)$$\Delta_{i}\left(t+K\right)=F_{s}\left(K\right)+F_{m}\left(K\right)f_{i}\left(t\right)\Delta_{i}\left(t\right)$$

We proceed with the max-weight policy derivation steps by the definition of Lyapunov function and Lyapunov drifts. Similar to [5], we use the quadratic Lyapunov function as given in Equation (Equation 21). However, rather than calculating the Lyapunov drift between consecutive frames, we calculate the Lyapunov drift $Y_{i}\left(t\right)$ between the current frame *t* and the expected query-arrival frame t+K. Calculation of Lyapunov drift is given in Equation (Equation 22).
(21)$$L\left(t\right)=\frac{1}{M}\sum_{i=1}^{M}\Delta_{q_{i}}^{2}\left(t\right)$$
(22)$$\begin{array}{cc} Y_{i}\left(t\right)=\; & E\left[\Delta_{i}^{2}\left(t+K\right)-\Delta_{i}^{2}\left(t\right)\right]\\ =\; & E\left[F_{s}{\left(K\right)}^{2}+2f_{i}\left(t\right)F_{s}\left(K\right)F_{m}\left(K\right)\Delta_{i}\left(t\right)+f_{i}\left(t\right)F_{m}^{2}\left(K\right)\Delta_{i}^{2}\left(t\right)-\Delta_{i}^{2}\left(t\right)\right]\\ =\; & E\left[F_{s}{\left(K\right)}^{2}-\Delta_{i}^{2}\left(t\right)+\left(1-a_{i}\left(t\right)c_{i}\left(t\right)\right)\left[2F_{s}\left(K\right)F_{m}\left(K\right)\Delta_{i}\left(t\right)+F_{m}^{2}\left(K\right)\Delta_{i}^{2}\left(t\right)\right]\right] \end{array}$$

In Equation (Equation 22), $a_{i}\left(t\right)$ is the only decision variable from which the scheduling policy can choose its value. For simplification, we ignore terms in the calculation of $Y_{i}\left(t\right)$ that are not affected by the decision $a_{i}\left(t\right)$. We denote the modifiable part of the Lyapunov drift with $a_{i}\left(t\right)$ decision as ${\hat{Y}}_{i}\left(t\right)$.
(23)$$\begin{array}{cc} {\hat{Y}}_{i}\left(t\right)= & -E\left[c_{i}\left(t\right)\right]E\left[a_{i}\left(t\right)\right]E\left[2F_{s}\left(K\right)F_{m}\left(K\right)\Delta_{i}\left(t\right)+F_{m}^{2}\left(K\right)\Delta_{i}^{2}\left(t\right)\right]\\ =\; & E\left[a_{i}\left(t\right)\right]p_{i}E\left[2F_{s}\left(K\right)F_{m}\left(K\right)\Delta_{i}\left(t\right)+F_{m}^{2}\left(K\right)\Delta_{i}^{2}\left(t\right)\right] \end{array}$$

At each frame, the main objective of the scheduling policy is to minimize the Lyapunov drift. Therefore, the scheduling policy must eliminate the receiver with the highest $C_{i}\left(t\right)$ to cause maximum reduction to Lyapunov drift.

**Lemma** **2.**
*For each frame, QAoI-aware max-weight (Q-MW) policy selects the receiver with highest immediate cost $C_{i}\left(t\right)$. Calculation of immediate cost is given in (Equation 24).*

(24)
$$C_{i}\left(t\right)=p_{i}F_{m}\left(K\right)\Delta_{i}\left(t\right)\left(F_{m}\left(K\right)\Delta_{i}\left(t\right)+2F_{s}\left(K\right)\right)$$



To emphasize what our system model corresponds to in practice and depict the difference between AoI- and QAoI-aware policies, we can consider a simple IoT network as an example. This network consists of sensors, microprocessors, a base station, and individual users. In the network, sensor devices generate time-sensitive data about their current status. Nevertheless, the sensors cannot process this data, and they have to transfer it over a wireless network to remote microprocessors. The sensors send the data to a base station, and the base station transmits this data over the wireless network. However, the transmission capacity of the base station is limited, and it cannot simultaneously transmit data to multiple processors.

There is a dedicated microprocessor for each sensor. Microprocessors use the sensor data and generate status reports. Each microprocessor is tracked by an individual user that queries the processor to obtain the freshest status report about the sensor. Query arrivals to each microprocessor are independent of each other and occur infrequently.

QAoI-aware policies come to the fore if the requirement in the system precedes the query source’s request for timely information. For the system model given in this example, AoI-aware policies concentrate on the AoI at the microprocessors, and the QAoI-aware policies focus on the AoI at the individual users. The impact of the QAoI-aware policy is shown in Figure 2. The figure examines the instantaneous AoI of a receiver (microcontroller in our example) in a multi-user network. A query source (individual user in our example) generates queries at the 41st, 81st, and 121st frames. From the query source’s perspective, freshness is only important at query instants. In line with the query source’s demands, the QAoI-aware policy aims to minimize the AoI of the receiver at the 41st, 81st, and 121st frames. Since there is no need for AoI minimization in all frames, the transmission constraint in the system can be relieved, and transmission attempts can be utilized more efficiently.

## 5. Implementation

In this section, we describe our implementation work on USRPs. We firstly share detailed information about the implementation environment. Then, we describe the packet interface that we use to transmit time-sensitive information in Section 5.2, and we explain the runtime of our setup in Section 5.3.

Software-based radios, also called "software radios" in pioneering studies, are radios that allow the user to change main parameters of communication systems such as center frequency, bandwidth, and coding of the communication system only by changing the software [32,33]. With SDRs, all layers of the communication system, from the physical layer to the application layer, can be changed only by software modifications. These radios play an important role in the development of today’s technologies that require rapid prototyping of various parameters, protocols, and standards, because software-based radios reduce the burden of extra hardware production for test and development studies and provide significant improvement in terms of time and cost.

For the AoI testbed implementation, we use one Ettus USRP N210, one NI USRP 2930, and two NI USRP 2930 SDR devices. General specifications of the devices are available in the devices’ datasheets [34,35]. Both USRPs have independent transmit and receive modules. For this reason, these devices can operate as a transmitter and a receiver simultaneously. Nevertheless, it is not possible to run two transmission operations simultaneously.

The host computer runs a LabVIEW application that interfaces with the USRP devices. USRP communicates with the host computer via a 1 Gb Ethernet link. Signals are fragmented to in-phase and quadrant components and carried over in the Ethernet packets. Each transmitter and receiver module contains an amplifier that is controllable through software. In the experiments, we often use these amplifiers to change channel reliabilities.

The LabVIEW environment contains useful built-in functions for system implementation. We use them frequently in our study. We also benefited from the examples regarding the PSK-modulated communication system and packet-based digital link tutorials and examples provided by LabVIEW and the LabVIEW community [36].

### 5.1. Setup

Among four USRPs, one USRP is configured as the base station, and the other three are the receiver modules. The setup configuration for the implementation is given in Table 1. An overview of the USRP testbed is given in Figure 3.

System time is discretized in 50 ms duration frames. The LabVIEW application keeps track of the frame number, that is, the total number of frames that have passed since the experiment began. The frame number is the system’s reference clock. All radios run in separate threads over a single LabVIEW application running on the host computer. Thus, difficulties related to synchronization are reduced, as all USRPs are managed from a single host.

We use QPSK modulation in the air interface. The maximum operating frequency of the USRP-2920 is 2.2 GHz [35], and we choose a center frequency of 1.9 GHz for all receivers. We prefer the high center frequency of the carrier signal to induce higher path loss since we have a limited area in the test environment.

The sampling rate of the USRP is configurable via the LabVIEW application. Detailed information about this configuration is described in the USRP documentation [35]. NI specifies that the I/Q ratio must be multiplied by 0.8 to convert to the sample rate [37]. In the implementation, we use the I/Q ratio 500k samples/s, which corresponds to a sampling rate of 400k samples/sec or bandwidth of 200 kHz. This bandwidth meets the requirements of our target application. Selecting higher I/Q rates increases the bandwidth. However, increasing the sample rate causes more data to be processed and transported. Therefore, more data would put a higher load on the USRP and Ethernet connection and eventually induce higher delay. Since timeliness is the primary concern in AoI calculations, we keep the I/Q Ratio low to achieve a more stable operating point without overloading the USRP and Ethernet.

### 5.2. Packet Interface

In the implementation, time-sensitive information is carried through the packets. The structure of the packet interface is summarized in Figure 4. There are six guard bits at the beginning of a packet. These bits are placed to prevent the pulse shaping filter from damaging the message content. The synchronization bit field starts after the guard bits. A 30-bit synchronization sequence is known in advance by both the sending and receiving modules. This sequence is created by a LabVIEW function that generates pseudo-random bits in the Galois domain. Receivers that continuously acquire data from the air interface use the synchronization sequence to detect the beginning of the packet.

The message field contains time-sensitive information. The message consists of Receiver ID (RX ID) and Packet ID fields. RX ID field is a 4-bit address that is used to distinguish receivers. Each receiver has a unique RX ID. When a receiver obtains a packet, it locates the RX ID field in the packet content and compares it with its RX ID. If the RX ID of the packet and the receiver do not match, the receiver discards the packet, and $\Delta_{i}\left(t\right)$ for that receiver increases by one for the next frame.

The frame number is the reference clock of the entire system. It initially starts from one at the beginning of the experiment and increases by one for each frame. Upon the generation of a packet, the Packet ID field is filled with the frame number of the system. Thus, the Packet ID field operates as the packet timestamp. Since the receiver also knows the current frame number, the difference between the packet’s creation frame (contained in the Packet ID field) and the current frame gives the instant information age $\Delta_{i}\left(t\right)$.

Packets sent over an unreliable channel may suffer corruption due to noise. The receiver should discard packets containing incorrect information since processing this data may lead to incorrect AoI measurements. To detect errors, we use cyclic redundancy check (CRC). Within the packet generation process, we pass the message field through the 16-bit CRC and write the result to the CRC field. When a receiver obtains a packet, it first calculates the CRC of the message field of the packet and compares the result with the CRC field in the packet. If both CRCs are equal, we consider the message to be error-free. We track the number of successful CRC checks for each receiver, thereby dynamically measuring the reliability of the channel. In the implementation, we dynamically estimate channel reliability throughout the experiment. Accurate calculation of the channel reliability values is essential, as MW policy and WIP take this value as input. We pre-run the setup to initialize the channel reliabilities. During the pre-run, we discard AoI calculations.

### 5.3. Runtime

The LabVIEW program allows multi-threading, which allows us to execute processes independently in different threads. We implement the Receiver, Transmitter, and Logging modules as separate threads in the program. In this way, we were able to perform these operations simultaneously. Moreover, the LabVIEW program has the feature of providing synchronization between threads. With the activation of this feature, it has been possible to organize processes running in different threads and following each other. The runtime of the system can be described step by step as follows:The new frame starts with incrementing the frame number.The scheduling policy performs the receiver selection for the new frame. AoIs of the receivers, query arrival status, and channel statistics are the inputs of the scheduling policy.In the meantime, receiver threads start acquiring a signal from receiver USRPs. The acquired signal is demodulated using LabVIEW’s built-in demodulation function. We synchronize the transmit and receive threads using the synchronization function of the LabVIEW program. Moreover, we keep the receiver thread’s acquisition duration long enough compared to the transmission thread’s time to complete its task so that the receiver can acquire the signal sent by the transmitter.A time-sensitive packet is constructed in the transmitter thread. The ID of the selected receiver is inserted into the new packet. The current frame number is also inserted in the “Package ID” field.The constructed packet is modulated using LabVIEW’s Modulation function and prepared for transmission. The transmitter thread transmits this packet to the USRP over an Ethernet connection.The transmitter USRP converts this packet to an RF signal and broadcasts it on the air interface.The receiver threads demodulate the signal in the air interface and try to catch the transmitted packet. The demodulator tries to detect synchronization bits to find the beginning of the packet. If the demodulator finds the synchronization bits, it returns a bit field that contains the packet.At the next stage, the receiver thread checks the CRC value of the acquired packet. The CRC field in the packet content is extracted. Next, the receiver thread passes the first 16 bits of the packet through the CRC. Then, the receiver compares this CRC result with the extracted CRC field in the packet. If both CRCs are equal, the receiver thread concludes that the packet is valid. Otherwise, the receiver discards the packet.The total number of successful CRC checks for each receiver is used to calculate channel reliabilities.After the CRC check, the receiver thread checks the Receiver ID field of the packet. If the Receiver ID field in the packet is different from the ID of the receiver, the receiver discards the packet again. If the Receiver IDs align, the acquired packet is assumed to be successfully received.Receiver threads that have finished their processes are set to idle for a while. The receiver thread will start listening to the air interface again before the new frame starts to activate the receiving process before the transmission occurs.Results obtained within a frame are passed to the logging thread. The main task of this thread is to calculate the average AoI, EAoI, and QAoI based on the result of the experiment. In addition, channel reliability and other statistics about the experiment are also calculated in this thread.Before the frame ends, the transmitter thread calculates instantaneous AoIs of the receivers with the data received within the frame. In the next frame, AoIs of the receivers and channel statistics will be used as input to the scheduling policy.

This experimental procedure is repeated at each frame. After the overall experiment is finished, results are saved to a text file.

## 6. Experiments and Results

Throughout this section, we share the results that we obtained in the USRP environment and MATLAB simulations. We share the performances of AoI-aware policies in Section 6.1, EAoI-aware policies in Section 6.2, and QAoI-aware policies in Section 6.3.

### 6.1. Evaluation of AoI-Aware Scheduling Policies

In this section, we share the results of the experiments conducted in the SDR network. We evaluate the performances of AoI-aware scheduling policies, and compare their AoI performances with round-robin and greedy policies. Round-robin policy activates all links sequentially, one per frame, regardless of any prior knowledge obtained about receivers. Greedy policy uses the AoIs of the receivers and selects the receiver with the highest age for packet transmission.

We evaluate the scheduling policies under various conditions by changing the channel reliabilities of the receivers among experiments. To change channel reliabilities, we manipulate the gains of the receiver and transmitter USRPs. LabVIEW allows configuring the signal gains of USRPs. Moreover, we locate receiver USRPs with different distances to the transmitter USRP to induce diverse path losses to receiver USRPs. When the signal power of the transmitter USRP increases, all receivers in the network acquire stronger signals. Therefore, the channel reliabilities of all receivers increase. Throughout the experiments, we also adjust the power gains of the receivers to alter the channel reliabilities. The receiver’s power gain is directly proportional to its channel reliability. Increasing the signal gain of a receiver reduces the error probability for that receiver and increases the channel reliability.

In the experiments, we run scheduling policies multiple times at each power gain level and take the average of the obtained results. We compare the scheduling policies in terms of the average AoI and the throughput of the network.

#### 6.1.1. Adjusting the Gain of an Individual Receiver

In this case, we increase the input signal gain of an individual receiver USRP. Throughout the experiments, we test the policies ten times at each transmitter gain level and average the results of redundant experiments. In each experiment, the frame length is K=7500, and M=3 receivers are available in the network. Results of the experiments in terms of AoI and throughput are given in Figure 5. Average channel reliabilities for each USRP gain level are given in Table 2.

#### 6.1.2. Adjusting the Gain of the Base Station

In this case, we increase the output signal gain of the transmitter USRP (base station). Throughout the experiments, we test the policies five times at each transmitter gain level and average the results of redundant experiments. In each experiment, the frame length is K=7500, and M=3 receivers are active in the network. Average AoI and throughput of age-aware policies are illustrated in Figure 6, and the channel statistics for the experimental setup are given in Table 3.

#### 6.1.3. Comparison of SDR Testbed Results with Simulations

In this section, we share the results of the comparison between simulation and realization. We use the results of the experiment mentioned in Section 6.1.2 as a reference to the simulation. We use the same channel reliabilities from Table 3 for the simulation environment and evaluate the policies. Results of the comparison in terms of average AoI and throughput are given in Figure 7.

#### 6.1.4. Interpretation of the Results

As channel reliability decreases, the performances of MW and WIP differ positively from the others. MW and WIP policies take channel reliability into account in the scheduling decision. This information enables more efficient use of transmission attempts. On the other hand, greedy policy does not utilize channel reliability information. If a receiver with a very low-quality channel is present in the network, the greedy policy may block the network by continuous unsuccessful update attempts to that receiver. This results in an increase in the average AoI of the network. For the first experiment set with the results illustrated in Figure 5, the performance degradation of the greedy algorithm is more apparent. Greedy policy always tries to send an update packet to the receiver with the worst channel condition. However, that receiver rarely receives packets successfully, and the base station gets stuck in that receiver until a successful packet reception. On the other hand, since the round-robin policy proceeds by transmitting to all receivers one by one without using any information about whether the packet is successfully received or not, the starvation problem does not occur. In both experiments, we also observe that as channel reliability values of receivers improve and asymmetry of channels decreases, greedy policy performs better than round-robin. As the channel conditions improve and the asymmetry among the channels decreases, performances of both policies converge to the optimal. As the channel reliability rises to 100%, all scheduling policies behave like round-robin and transmit to all receivers in a cyclic order.

For the SDR testbed simulation comparison case, we use the same average channel reliabilities in both experiments. We do not observe any significant difference in throughput, as expected. However, in terms of AoI, we found that the simulation results yield lower AoI than the SDR implementation. In the simulation environment, the channel status is a Bernoulli random variable. However, in the SDR implementation, the channel status is formed by realistic conditions and doesn’t have to be stationary or follow the Bernoulli distribution. The regularity of the packet arrivals is an essential factor for low AoI. Even if the channel reliabilities over time are equal for SDR realization and simulation environments, the imperfections of the realistic channel may reduce the update regularity more drastically than the Bernoulli-distributed channel.

### 6.2. Evaluation of EAoI-Aware Scheduling Policies

In this section, we compare the EAoI-aware policies with the traditional policies. Traditional policies do not utilize query statistics for scheduling decisions, and we aim to observe the outcomes of using query statistics. We evaluate the policies in the SDR environment and use EAoI as the primary performance indicator. We also share results about AoI and throughput metrics. Throughout the experiments, query presences at each frame are implemented as i.i.d. Bernoulli random variables. In each experiment, the frame length is K=7500, and M=3 receivers are active in the network. We use the proactive serving method as the query response scenario.

#### 6.2.1. Adjusting the Gain of an Individual Receiver

In this case, we increase the output signal gain of an individual receiver USRP. Throughout the experiments, we test the policies ten times for each gain level and average the results of redundant experiments. Evaluation of EAoI and AoI throughout the experiments are given in Figure 8. Channel statistics corresponding to USRP gain levels are given in Table 4 and query statistics are given in Table 5.

#### 6.2.2. Adjusting the Gain of the Base Station

In this case, we increase the output signal gain of the transmitter USRP (base station). Throughout the experiments, we test the policies at least five times for each gain level and average the results of redundant experiments. Evaluation of EAoI and AoI throughout the experiments are given in Figure 9. Channel statistics corresponding to USRP gain levels are given in Table 6 and query statistics are given in Table 7.

#### 6.2.3. Interpretation of the Results

For the EAoI minimization objective, the EAoI-MW and EAoI-WI policies outperform the policies that do not utilize query information. Moreover, experimental results show that EAoI-MW surpasses the EAoI-WIP in terms of throughput. For EAoI-aware scheduling policies, whether the policy is derived for the instantaneous serving or the proactive serving scenario does not cause a significant difference in EAoI performance. Rather than utilizing the exact timings of the query arrivals, EAoI-aware policies weight receivers according to their long-term query arrival statistics. Since there is no significant difference between the proactive response and instant response scenarios in the long-term query arrival statistics, there is no significant difference between the performances of the policies. As can be seen from Figure 8 and Figure 9, the EAoI performances of EAoI-MW derived for the instantaneous response scenario and the EAoI-WIP derived for the proactive response scenario are very close to each other.

### 6.3. Evaluation of QAoI-Aware Scheduling Policies

In this section, we share the results of our experiments. We conducted the experiments in the simulation environment and the SDR environment. Throughout the experiments, we evaluated the performance of the QAoI-aware MW policy in terms of QAoI and AoI, and we used the AoI-aware MW policy as a benchmark.

#### 6.3.1. Results from Simulation Environment

We conducted four experiments in the MATLAB environment. In each experiment, the frame length was K= 1,100,000, and M=10 receivers were active in the network. Within the experiments, we adjusted the query period of the receivers and observed the result of this increment from the AoI and QAoI perspectives. We initialized query periods to prevent the overlap of the query frames for each receiver. We assume $a_{i}\left(t\right)$ is stationary through time by taking advantage of non-overlapping queries, and we calculate $f_{i}$ as $f_{i}=1-p_{i}$ in Q-MW policy. Channel reliabilities (long-term average packet success rates) measured in the experiments are summarized in Table 8. Average QAoI and AoI obtained by Q-MW policy for each experiment are given in Figure 10 and Figure 11, respectively.

#### 6.3.2. Results from the USRP Testbed

We conducted two experiment sets in the SDR testbed. In both experiment sets, we increased the output signal gain of the transmitter USRP (base station) to observe the effects of various channel reliabilities (i.e., packet success rates). For each signal gain level, we test the policies at least ten times and average the results of redundant experiments. The frame length of each test was K=7500, and there were M=3 receivers in the network. In the first experiment set, the query period of receivers was 25, and in the second experiment set, the query period of the receivers was 5. In both experiment sets, we initialize the query periods to prevent the arrival of multiple queries within the same frame. We assume $a_{i}\left(t\right)$ is stationary through time by taking advantage of non-overlapping queries, and we calculate $f_{i}$ as $f_{i}=1-p_{i}$ in Q-MW policy.

For the first experiment set, evaluation of QAoI and AoI are given in Figure 12 and Figure 13, respectively. Channel reliabilities corresponding to USRP gain levels are given in Table 9. For the second experiment set, evaluation of QAoI and AoI are given in Figure 14 and Figure 15, respectively. Channel reliabilities corresponding to USRP gain levels are given in Table 10.

In the second experiment, we increase the output signal gain of the transmitter USRP (base station). Throughout the experiments, we test the policies at least ten times for each gain level and average the results of redundant experiments. In each experiment, the frame length is K=7500, and M=3 receivers are active in the network. In this experiment, the query period for each receiver is 5 frames. We initialize the query periods such that the queried frames of receivers do not overlap at the same frame.

#### 6.3.3. Comparison of SDR Testbed Results with Simulations

In this section, we share the results of the comparison between simulation and realization. We use the results of the experiment illustrated in Figure 14 as a reference for the simulation. We use the same channel reliabilities from Table 10 for the simulation environment. Results of the comparison in terms of QAoI are given in Figure 16.

#### 6.3.4. Interpretation of the Results

Within the scope of the experiments, we studied the case where query arrivals are periodic. According to the results of both SDR realization and simulations, the Q-MW policy outperforms the AoI-MW policy for the QAoI minimization objective. By utilizing the query arrival information, the Q-MW scheduling policy can select receivers more efficiently, and thus it can exhibit superior QAoI performance compared to AoI-MW.

Throughout the simulations, we investigated Q-MW in networks with various channel reliabilities. In the first experiment, we considered ten receivers with good channel reliability. According to the results of this experiment, in cases where the query periods of the receivers do not overlap, the QAoI policy can reduce the average QAoI in the network to approximately one, which is the lowest possible limit. In the first experiment, the expected number of attempts to update a receiver is close to one. Having a reduced number of attempts enables the scheduling policy to distribute scheduling decisions more effectively and eases the alignment of the scheduling decisions with the query periods. In the second experiment, all receivers have poor channel qualities, and the number of attempts needed to update a receiver is high. As the channel reliabilities decrease, the expected number of attempts to update a receiver increases, and aligning the scheduling decisions with the query periods become more challenging. In this case, the performance of query-aware policies is reduced. As the query period increases, queried frames of the receivers become more distant from each other, which positively affects the performance of query-aware policies.

The fact that the transmission can only be allocated to a single receiver in each frame is one of the most fundamental limitations of the network. The QAoI-aware MW policy we recommend, on the other hand, reduces the need for packet transmission in the network by taking into account the query periods of receivers’ timely information requests and eases the transmission allocation constraint in line with the query periods.

## 7. Conclusions and Future Work

Within the scope of the paper, we have examined the AoI, EAoI, and QAoI, which are semantic communication metrics that prioritize information freshness. We implemented a multi-user wireless network with SDRs to examine these metrics in real-world scenarios. We investigated the performance of AoI-aware scheduling policies by comparing their AoI performance with traditional scheduling policies. The emulation results reveal that the WI and MW policies are superior to the round-robin and greedy policies, as they exploit the information on the link reliabilities and AoIs of the receivers. Experimental results obtained in the SDR testbed are close to simulation results when packet drops are rare, but as the link reliabilities decrease, they begin to show some slight discrepancies. We attribute this to the following: the AoI-aware policies adopted in this work were derived under Bernoulli-distributed packet drops; however, as channels get poorer, the sequence of packet drops tends to acquire a memory.

We have also studied the Effective AoI and Query AoI metrics to examine the freshness of information from the perspective of the query source in pull-based networks. For the EAoI domain, we proposed EAoI-MW policy by leveraging the formerly defined AoI-aware policies. We implemented and tested the EAoI-MW and EAoI-WI policies on the SDR Network. Experiment results show that utilizing the statistical information about the query process significantly improves EAoI performance. We observed that EAoI-MW policy exhibits a comparable performance with EAoI-WI and yields better results throughput. For the Query AoI metric, we have proposed a scheduling policy by adapting the max-weight policy to the QAoI case for multi-user pull-based network scenarios. We tested the resulting Q-MW policy in simulation and SDR implementation environments. Results reveal that utilizing the Q-MW policy can reduce QAoI significantly compared to AoI-aware policies.

In future studies, we seek to examine different semantic metrics beyond AoI. To this end, we want to expand the scope of our work on the QAoI. We also aim to investigate and optimize the Q-MW policy for the stochastic query arrival scenarios.

## Figures and Tables

**Figure 1 entropy-24-00673-f001:**
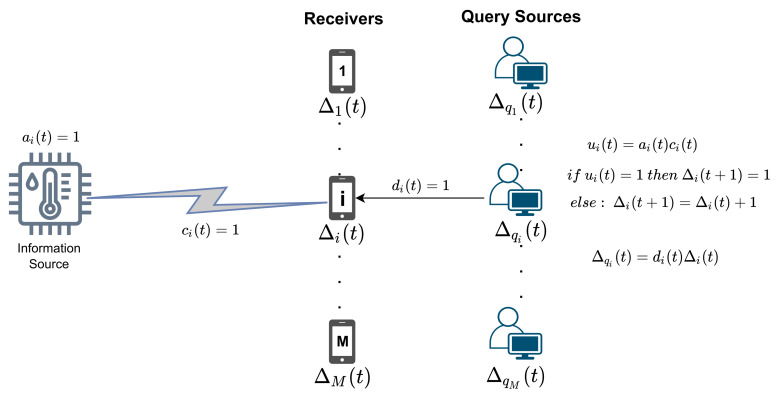
The architecture of the system model.

**Figure 2 entropy-24-00673-f002:**
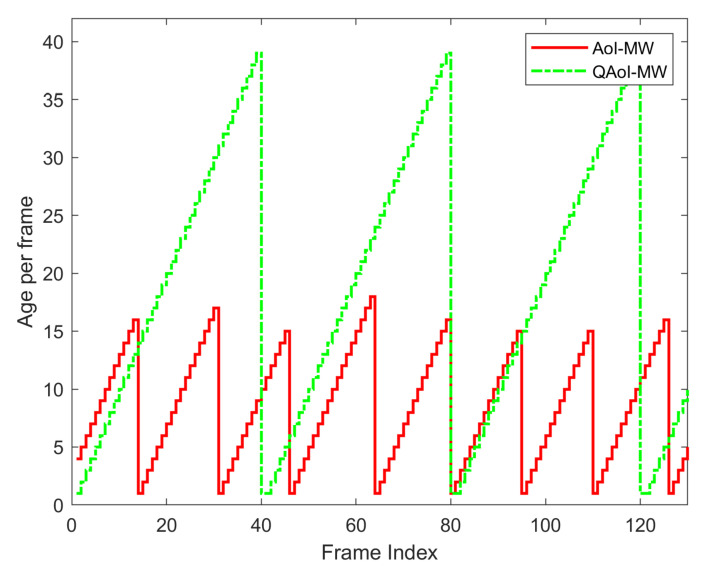
Instantaneous AoI of a receiver in a multi-user network.

**Figure 3 entropy-24-00673-f003:**
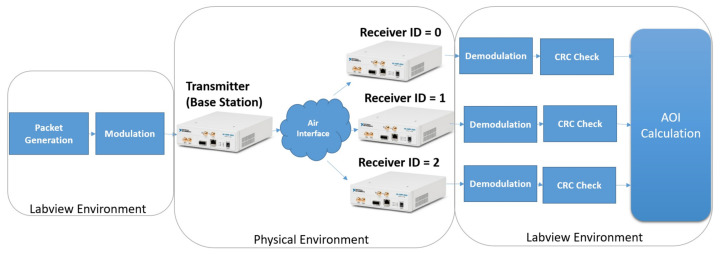
Overview of the implementation environment.

**Figure 4 entropy-24-00673-f004:**
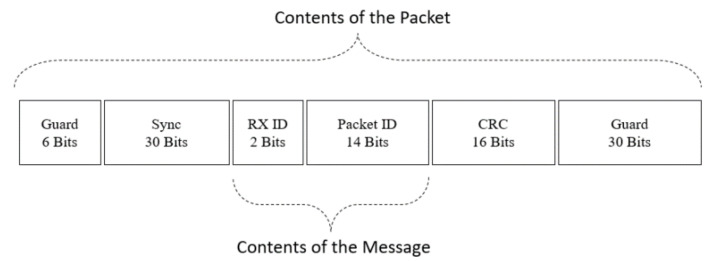
Packet content in the air interface.

**Figure 5 entropy-24-00673-f005:**
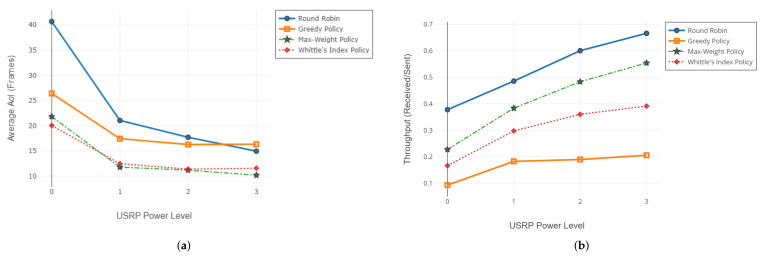
Evaluation of average AoI $J_{A}$ (**a**) and throughput (**b**) with varying receiver gain (SDR testbed).

**Figure 6 entropy-24-00673-f006:**
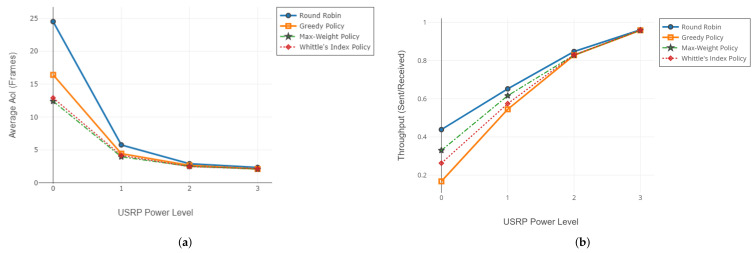
Evaluation of average AoI $J_{A}$ (**a**) and throughput (**b**) with varying BS output gain (SDR testbed).

**Figure 7 entropy-24-00673-f007:**
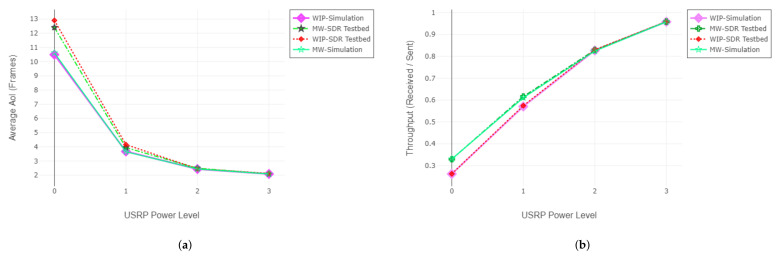
Comparison of simulation and implementation in terms of average AoI $J_{A}$ (**a**) and throughput (**b**).

**Figure 8 entropy-24-00673-f008:**
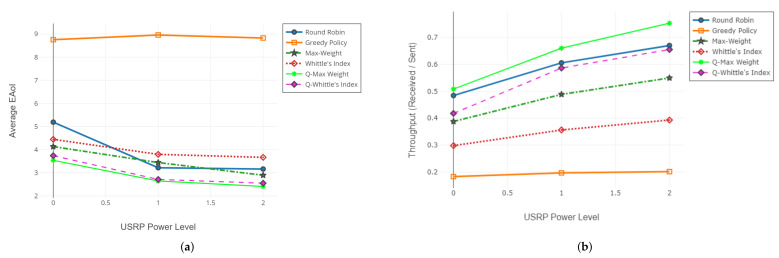
Evaluation of effective AoI $J_{E}$ (**a**) and throughput (**b**) with varying input gain of second receiver (SDR testbed).

**Figure 9 entropy-24-00673-f009:**
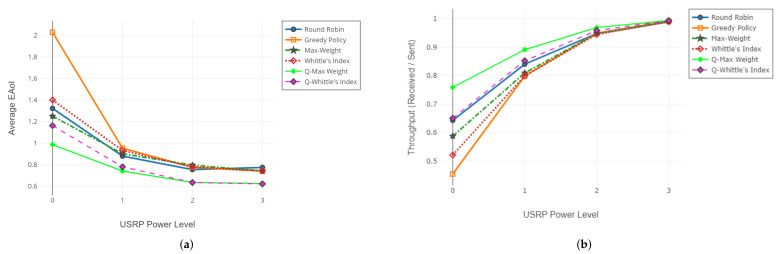
Evaluation of effective AoI $J_{E}$ (**a**) and throughput (**b**) with varying output gain of BS (SDR testbed).

**Figure 10 entropy-24-00673-f010:**
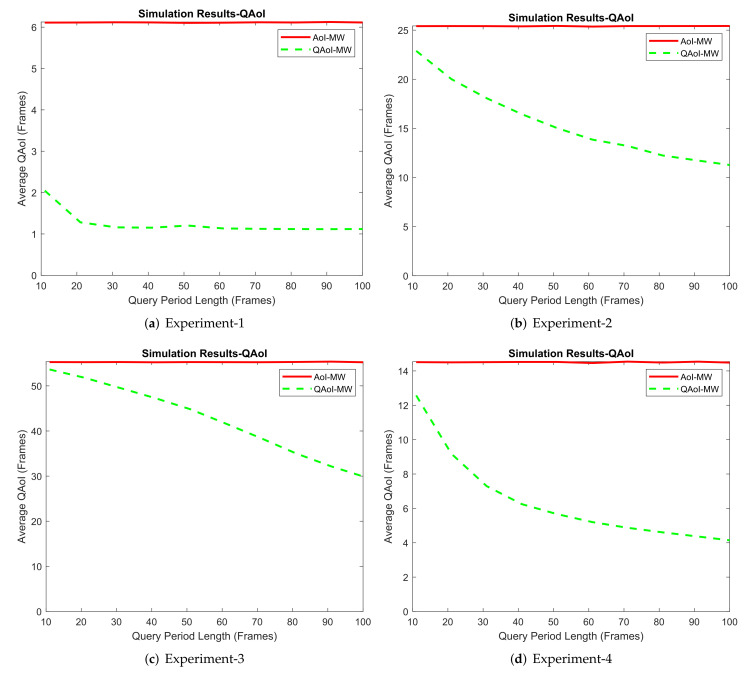
Evaluation of Q-MW policy in terms of average QAoI ($J_{Q}$).

**Figure 11 entropy-24-00673-f011:**
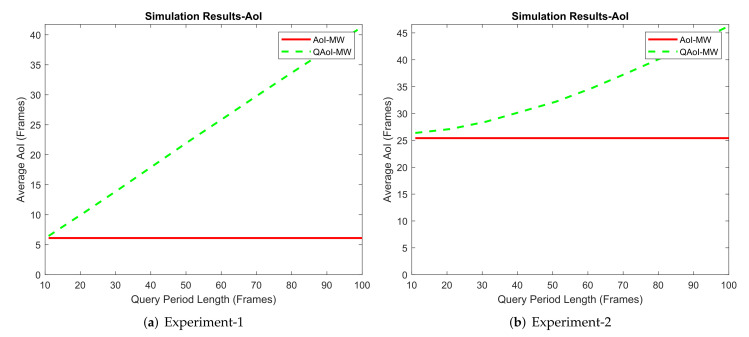
Evaluation of Q-MW policy in terms of average AoI ($J_{A}$).

**Figure 12 entropy-24-00673-f012:**
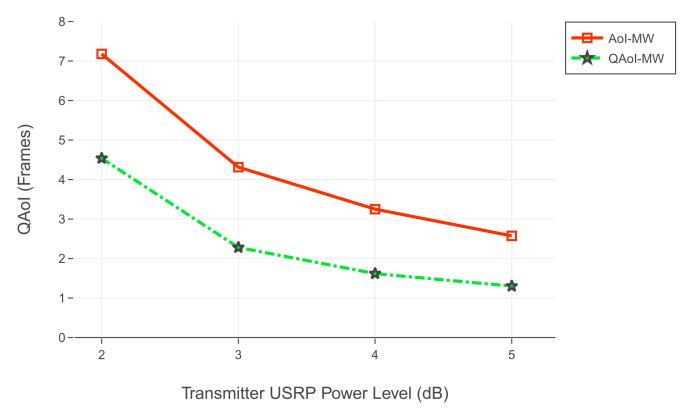
Evaluation of QAoI ($J_{Q}$) for varying power levels of transmitter USRP, 25 frames length query period.

**Figure 13 entropy-24-00673-f013:**
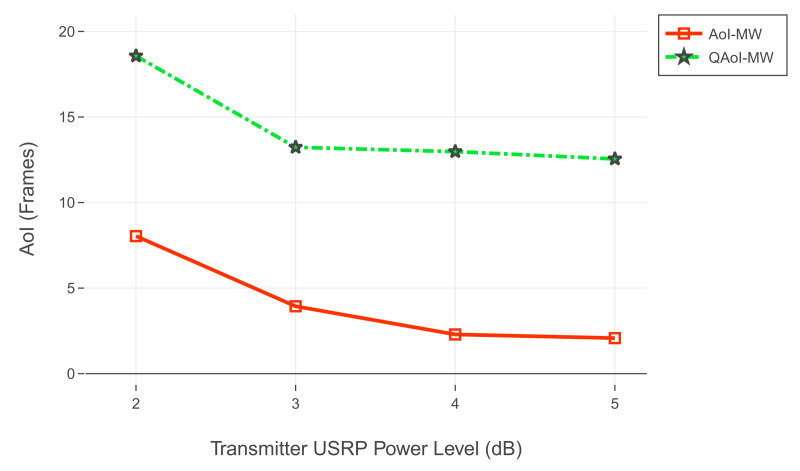
Evaluation of AoI ($J_{A}$) for varying power levels of transmitter USRP, 25 frames length query period.

**Figure 14 entropy-24-00673-f014:**
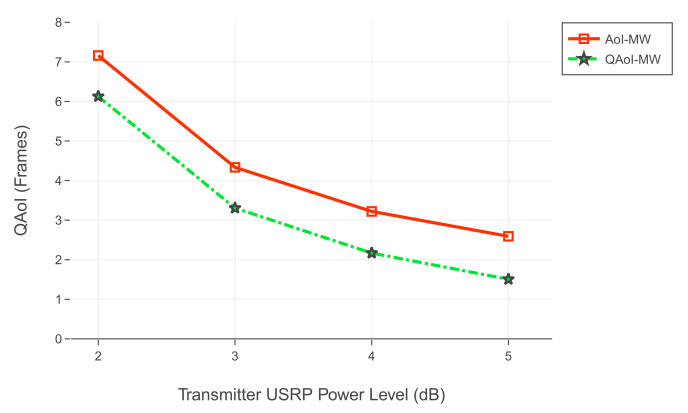
Evaluation of QAoI for varying power levels of transmitter USRP.

**Figure 15 entropy-24-00673-f015:**
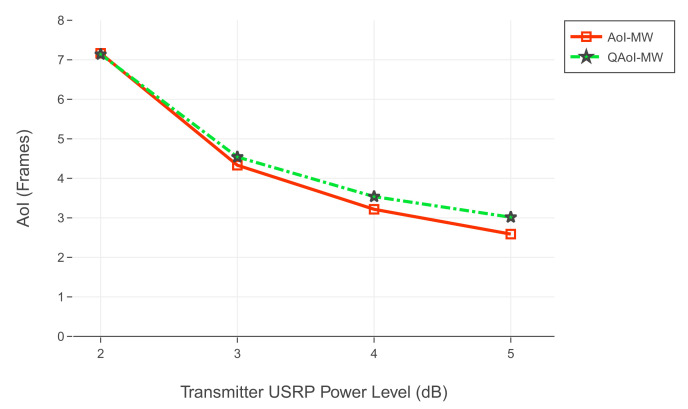
Evaluation of AoI for varying power levels of transmitter USRP.

**Figure 16 entropy-24-00673-f016:**
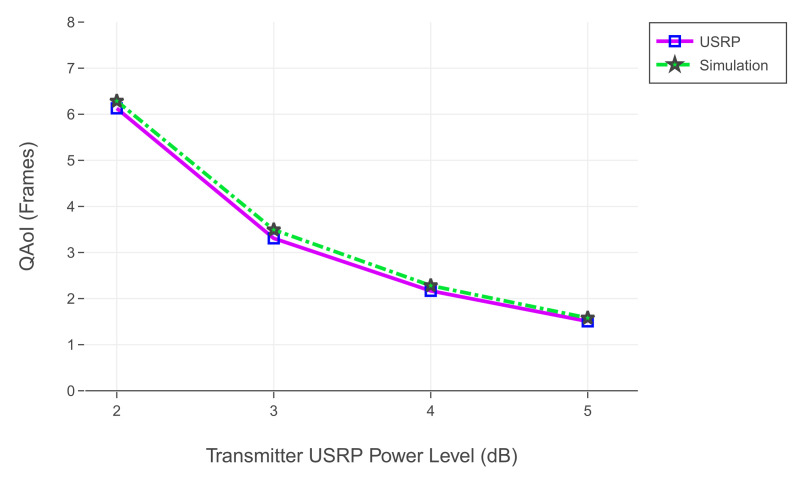
Comparison of simulation and realization results.

**Table 1 entropy-24-00673-t001:** Overview of parameters.

Modulation:	Quadrature Phase-Shift Keying (QPSK)
Center Frequency:	1.9 GHz
I/Q Rate:	500k Samples/s
Sample Rate:	400k Samples/s
Bandwidth:	200 kHz
Bits Per Symbol:	2
Samples Per Symbol:	8
Duration of one Frame:	50 ms

**Table 2 entropy-24-00673-t002:** Channel statistics in the first experiment set.

Experiment Index	$p_{1}$	$p_{2}$	$p_{3}$	Coefficient of Variation $C_{V}$ among Channels
0	0.9997	0.0517	0.0779	0.84
1	0.9997	0.3698	0.078	1.16
2	0.9997	0.7135	0.0747	1.337
3	0.9997	0.9139	0.0795	1.361

**Table 3 entropy-24-00673-t003:** Channel statistics in the second experiment set.

Experiment Index	$p_{1}$	$p_{2}$	$p_{3}$	Coefficient of Variation $C_{V}$ among Channels
0	0.9997	0.0814	0.2317	0.988
1	0.9997	0.3566	0.5891	0.496
2	0.9997	0.6811	0.8587	0.196
3	0.9997	0.9055	0.9733	0.051

**Table 4 entropy-24-00673-t004:** Channel statistics.

Gain	$p_{1}$	$p_{2}$	$p_{3}$
0	0.999	0.363	0.078
1	0.999	0.735	0.076
2	0.999	0.930	0.077

**Table 5 entropy-24-00673-t005:** Query statistics.

$q_{1}$	$q_{2}$	$q_{3}$
0.9	0.9	0.1

**Table 6 entropy-24-00673-t006:** Channel statistics.

Gain	$p_{1}$	$p_{2}$	$p_{3}$
0	0.9997	0.6652	0.2439
1	0.9997	0.9073	0.5983
2	0.9997	0.9830	0.8651
3	0.9997	0.9980	0.9736

**Table 7 entropy-24-00673-t007:** Query statistics.

$q_{1}$	$q_{2}$	$q_{3}$
0.9	0.1	0.1

**Table 8 entropy-24-00673-t008:** Channel statistics for simulations.

Experiment Index	$p_{1}$	$p_{2}$	$p_{3}$	$p_{4}$	$p_{5}$	$p_{6}$	$p_{7}$	$p_{8}$	$p_{9}$	$p_{10}$
1	0.9	0.9	0.9	0.9	0.9	0.9	0.9	0.9	0.9	0.9
2	0.9	0.9	0.9	0.9	0.9	0.1	0.1	0.1	0.1	0.1
3	0.1	0.1	0.1	0.1	0.1	0.1	0.1	0.1	0.1	0.1
4	0.1	0.2	0.3	0.4	0.5	0.6	0.7	0.8	0.9	1

**Table 9 entropy-24-00673-t009:** Channel statistics.

USRP Power Level	$p_{1}$	$p_{2}$	$p_{3}$
2	0.9997	0.2331	0.1681
3	0.9997	0.4203	0.3175
4	0.9997	0.5567	0.4775
5	0.9997	0.7245	0.6589

**Table 10 entropy-24-00673-t010:** Channel statistics.

USRP Power Level	$p_{1}$	$p_{2}$	$p_{3}$
2	0.9997	0.2380	0.1687
3	0.9997	0.4109	0.3167
4	0.9997	0.5721	0.4796
5	0.9996	0.7299	0.6635

## Data Availability

Not applicable.

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
