# Peer review of "Implementation and Evaluation of Age-Aware Downlink Scheduling Policies in Push-Based and Pull-Based Communication"

_entropy, 2022, doi:10.3390/e24050673_

Round 1

Reviewer 1 Report

The author describes the results of a validation of scheduling policies to ensure high network throughput in fifth-generation (5G) communications. Through a meticulous validation, the authors found that using a Q-MW policy significantly reduces QAoI compared to using an AoI-aware policy. The reviewers agree to publish this paper.

However, please correct the grammatical errors listed below.

L.21

"are" instead of "is".

L.23

"and" is duplicated.

L.165

Please change the subject from the reference number to the name of the author of the cited paper. Then put the reference number in the appropriate position in the sentence.

[31] investigates

Two lines below L.215

There is a misspelling.

Wrong: "tranmission"

Correct: "transmission"

L.254

Delete "comma ," after "Because".

L.258

Insert a "comma ," after "Therefore,".

Between L.295 and L.296 The quotation in equation (15) is not bracketed.

Please change it.

Wrong: The result is given in 15.

Correct: The result is given in (15).

L.443

The subject is plural.

Wrong: increases.

Correct: increase

L.451

There is a misspelling.

Wrong: indivividual

Correct: individual

Author Response

We would like to sincerely thank the reviewer for very careful reading of the manuscript, and constructive comments. We have revised the manuscript, and we believe we have addressed all the issues raised by the reviewers. Please see the attachment

Reviewer 2 Report

The manuscript presents the scheduling policies in push/pull communication. It is well presented and organized. However, it contains some drawbacks:

  1. the Introduction is too long !!
  2. the contribution of the manuscript is not defined
  3. related works section has to be added
  4. the architecture of the proposed model has to be presented in graphical form
  5. please start each section with the sort introduction
  6. add the practical examples of the use of your approach/policy - gove the practical exmples when you can use it, explain why what what it gives
  7. figure 1 is too small - not visible at all !!!; the caption for figure 1 is too long - it has to be 1 sentence !!!
  8. there is no reference to figure 1 in the text
  9.  

    the same for figure 2
  10. there is no reference to table 1 in the text
  11. figures 3-7 are too small, not visible

Author Response

We would like to sincerely thank the reviewers for their very careful reading of the manuscript, and their constructive comments. We have revised the manuscript, and we believe we have addressed all the issues raised by the reviewers. In addition to the changes requested by the reviewers, we have also corrected some typos and improved the presentation of the paper. Please see the attachment.

Reviewer 3 Report

"Implementation and Evaluation of Age-Aware Downlink Scheduling Policies in Push-Based and Pull-Based Communication" is a very interesting manuscript dealing with a subject of great importance and extremely current.
Technically and Scientifically sound, and written in a clear, detailed and rigorous way, with very interesting results, undoubtedly deserves to be published in Entropy.
The Introduction works here as a supporting pillar for the rest of the text. It has a remarkable literature review. It lacks a single paragraph at the end describing the article's structure.
Sections 2. System Model and 3. Age-Aware Downlink Scheduling Policies are very good.
In section 4. Implementation, the procedure described in subsection
4.3. Runtime must be presented in the form of divided into sequenced and numbered steps, in order to facilitate its understanding.
Sections 5. Experiments and Results and 6. Conclusions and Future Work are ok. In Figure 1 and Figure 2, titles are not distinguished from comments. It has to be fixed.
For example "[31] investigates", in line 164 is incorrect: a paragraph or a sentence must not begin with [ ]. Use, for example, In [31] the authors investigate...; check other similar situations.

Author Response

(The authors gave the same response as above.)

Round 2

Reviewer 2 Report

Still the manuscript contains some drawbacks:

  1. the Introduction is too long
  2. the presentation of the papre's structure has t be given at the of Intrduction, NOT related works !!!!!
  3. related works section has to be given as separate section
  4. the architecture of the proposed model has to be presented in graphical form
  5. add the practical examples of the use of your approach/policy - gove the practical exmples when you can use it, explain why what what it give

Author Response

We would like to sincerely thank the reviewer for the constructive comments. We have revised the manuscript, and we believe we have addressed all the issues raised by the reviewer.
